# Higher Yield and Polyphenol Content in Olive Pomace Extracts Using 2-Methyloxolane as Bio-Based Solvent

**DOI:** 10.3390/foods11091357

**Published:** 2022-05-07

**Authors:** Christian Cravotto, Anne Sylvie Fabiano-Tixier, Ombéline Claux, Vincent Rapinel, Valérie Tomao, Panagiotis Stathopoulos, Alexios Leandros Skaltsounis, Silvia Tabasso, Laurence Jacques, Farid Chemat

**Affiliations:** 1GREEN Extraction Team, INRAE, UMR 408, Avignon Université, F-84000 Avignon, France; christian.cravotto@alumni.univ-avignon.fr (C.C.); ombeline.claux@univ-avignon.fr (O.C.); farid.chemat@univ-avignon.fr (F.C.); 2Pennakem Europa, 224 Avenue de la Dordogne, F-59640 Dunkerque, France; vincent.rapinel@ecoxtract.com (V.R.); laurence.jacques@minakem.com (L.J.); 3MicroNut Team, INRAE, UMR 408, Avignon Université, F-84000 Avignon, France; valerie.tomao@univ-avignon.fr; 4Department of Pharmacognosy, School of Pharmacy, University of Athens, Panepistimioupolis, 15571 Zografou, Greece; stathopoulos@pharm.uoa.gr (P.S.); skaltsounis@pharm.uoa.gr (A.L.S.); 5Department of Chemistry, University of Turin, Via P. Giuria 7, 10125 Turin, Italy; silvia.tabasso@unito.it

**Keywords:** 2-methyloxolane, hexane, olive pomace, polyphenols

## Abstract

Despite its severe toxicity and negative environmental impact, hexane remain the solvent of choice for the extraction of vegetable oils. This is in contrast with the constantly growing demand for sustainable and green extraction processes. In recent years a variety of alternatives to hexane have been reported, among them 2-methyloxolane (2-MeOx), which has emerged as a promising bio-based alternative. This study evaluates the possibility of replacing hexane, in the extraction of olive pomace (OP), with 2-MeOx, both dry and saturated with water (4.5%), the latter of which is called 2-MeOx 95.5%. The three solvents have been compared in terms of extraction yield and quality, as well as the lipid and polyphenol profiles of the extracts. The work concluded that both dry 2-MeOx and 2-MeOx 95.5% can replace hexane in OP extraction, resulting in higher yields and extracts richer in phenolic compounds. This study should open the road to further semi-industrial scale investigations toward more sustainable production processes.

## 1. Introduction

In the period 2017–2020, worldwide olive oil production was over 3 million tons per year with the European Union being the leading producer, consumer and exporter of olive oil. Total European production was around 2 million tons of oil per year (68% of the world’s olive oil production) and the main producers of the member States are Spain (66% of EU production), Italy (15%) and Greece (13%), while European consumption is about 1.5 million tons (50% of world production) [1].

Olive oil is mostly found in the form of small drops in the vacuoles of mesocarp cells. Historically the olive oil was recovered from the fruit by mechanical press using old methods but nowadays oil recovery is performed in modern mills that have increased production efficiency while maintaining high product-quality standards [2]. The olive oil production process is presented in Figure 1. After the leaves are separated, the olive fruits are washed with water to remove all contaminants. The crushing process is the first step, and this is followed by malaxation in which a horizontal shaft with rotating arms and blades slowly mixes the paste for about 30–40 min in order to increase oil yield by facilitating the coalescence of small oil droplets [3].

In the last step, decanter centrifugation separates the extra-virgin (EVOO) and virgin (VOO) olive oil from the olive pomace (OP) (pulp and stones) and wastewater, thanks to their different densities. The difference between VOOs and EVOOs is the maximum permitted acidity index, expressed as g of oleic acid (OLA). In VOOs the acidity must not exceed 2 g of OLA per 100 g of oil, while for EVOOs it must not exceed 0.8 g of OLA per 100 g of oil. In the two-phase process, olive oil is separated from the wet OP, which has a water content of about 70% by weight, while, in the three-phase process, OP and wastewater are recovered separately [4]. These two by-products show a remarkable content of bioactive compounds within particular polyphenols, with well-recognized benefits for human health, including the prevention of cardiovascular diseases, cancers, and osteoporosis [5,6,7,8]. In addition to their important pharmaceutical and nutraceutical uses, the cosmetic applications of polyphenols are emerging thanks to their anti-aging, antimicrobial, anti-inflammatory, and photoprotective actions [9].

Each ton of olives roughly generates 0.5–0.6 tons of OP and a huge amount of olive mill wastewater is generated, in the range of 0.2–0.5 m^3^ per ton of olives, in the three-phase centrifugation system [10]. Although three-step processes give a slightly higher oil yield, two-step processes are usually preferred due to reduced water requirements with the added advantage of not diluting the water-soluble olive oil phenols. 

Among the well-known phenolic compounds in olive oil and OP we can find hydroxytyrosol (HT), tyrosol (T) and oleuropein (OL), and these are considered to be the main causes for the beneficial effect on human health. Furthermore, EFSA (European Food Safety Authority) approved a health claim in 2011 related to the protection of blood lipids from oxidative stress by phenolic compounds contained in olive oil. The claim can only be used for olive oil containing at least 5 mg of HT and its derivatives (e.g., OL complex and T) per 20 g of oil. The consumer must be informed that the healthy effect requires a daily intake of at least 20 g of olive oil [11]. In addition, olive oil contains high concentrations of unsaturated fatty acids, mainly oleic acid (approximately 70% of total), and can therefore benefit from an additional EFSA-approved health claim that correlates the maintenance of normal blood cholesterol levels with replacing saturated with unsaturated fats in a person’s diet.

In accordance with circular economy principles, OP is a by-product that is rich in bioactive compounds and that can be valorised through the recovery of residual oil and antioxidant compounds, mainly polyphenols, that are largely lost at the end of the olive oil production process [12,13]. While a significant effort has been made to identify and recover the phenolic compounds contained in OP, including with the use of new technologies, ways of improving OP oil recovery have not been properly studied, to the best of our knowledge, and this process is still performed using conventional methods and solvents [14,15,16,17,18,19,20,21].

Approximately 12–15% of total olive oil content is still present in OP and is normally recovered by hexane extraction, after matrix drying [22]. The crude OP oil then requires chemical refining to remove undesirable compounds, such as free fatty acids, phospholipids, colours, etc. Finally, the refined OP oil is mixed with a proper amount of virgin olive oil (normally around 10%), thus resulting in commercially edible OP oil [23], which is a lower quality product compared to EVOOs and VOOs.

Hexane belongs to the family of volatile organic compounds, commonly known as VOCs, and is currently the most frequently used solvent for the extraction of vegetable oils [24]. Growing concern about hexane toxicity, environmental pollution and sourcing from non-renewable resources has prompted the European Union to introduce stricter regulations and classify it as a neurotoxic and suspected reprotoxic substance (REACH) [25,26]. In recent years, 2-methyltetrahydrofuran, also known as 2-methyloxolane (2-MeOx), has emerged as a valid alternative to hexane in oil extraction, thanks to his high affinity for lipids, a boiling point (80 °C) that is compatible with the existing extraction process, bio-based sourcing and safer toxicological profile (comparison with *n*-hexane in Figure 2) [27].

Several solvents have initially been investigated using COSMO-RS, which is quantum chemistry-based software that predicted that 2-MeOx may be comparable to *n*-hexane at solubilising lipids and even superior in case of more polar lipids (e.g., phospholipids) and other secondary metabolites, such as phenolic compounds [24,36].

Theoretical calculations have been supported by several studies that have shown that the use of 2-MeOx, compared to hexane, results in increased extraction yield and higher polar lipid and polyphenol content in the extracts from both plant and animal sources [37,38,39].

On an industrial scale, the solvent recycling process is a crucial step and, when using 2-MeOx, after distillation and decanter separation, the recovered organic phase is saturated with water. The mixture of 2-MeOx/H_2_O (95.5/4.5% at 55 °C) must be sent to an additional distillation step to recover dry 2-MeOx. This second distillation requires a lot of energy, for that reason, the extractive efficiency of 2-MeOx 95.5% has also been studied to assess whether it would be more convenient to bypass the additional distillation step using the saturated organic phase directly.

This study aims to evaluate the ability of 2-MeOx (both dry and 95.5%) to extract OP by comparing it with hexane in terms of yield, quality and extract composition, see Figure 3.

## 2. Materials and Methods

**COSMO-RS Evaluation**. The conductor-like screening model for real solvents (COSMO-RS) was selected to predict the solubilities of the target solutes that are commonly found in olive oil and olive pomace in *n*-hexane, 2-MeOx and 2-MeOx 95.5%. COSMO-RS is a calculation method developed by Klamt and is based on a quantum chemistry model, and has been applied over the last two decades for solubility prediction in the context of natural product extraction [40]. The COSMOThermX software (18.0.2, COSMOlogic version GmbH & Co., Leverkusen, Germany) was used for calculations at 25 °C, referring to “iterative” mode, absolute calculations, and took solutes and solvents to be in the pure and liquid state. The results are expressed as log10 (x_solub_), the logarithm of the molar fraction of solute in the solvent. The closer the value of log10 (x_solub_) is to zero, the greater the predicted solubility of the solute.

**Solubility measurements**. The solubilities of hydroxytyrosol (HT), tyrosol (T), caffeic acid (CA) and *p*-coumaric acid (C) were determined at room temperature (25 °C) with high purity standards (96–99%) in *n*-hexane, dry 2-MeOx and 2-MeOx 95.5%. Saturated solutions were stirred for two hours with a rotator (ClearLine Dominique Dutscher, Bernolsheim, France) and then kept still for 24 h. Saturated solution supernatants were then filtered with syringe filters (0.20 µm) and analysed using UPLC-DAD (Waters, Milford, MA, USA). Each experimental value is an average of at least three different measurements and expressed as g/L [41].

**Ternary phase diagram of olive oil/2-MeOx/water**. The ternary phase diagrams of olive oil, 2-MeOx and water were implemented at 55 °C (conventional temperature in industrial extractions), to assess whether 2-MeOx 95.5% could also be a good solvent in olive oil extraction. Appropriate amounts of dry 2-MeOx and olive oil (extra-virgin olive oil quality) were accurately weighed into glass tubes and solutions with different percentages of the two liquids being obtained. The tubes were then closed and placed in a temperature-controlled water bath at 55 °C. Water was then added drop wise to the solution until turbidity appeared. The amount of water added was then established by re-weighing the tubes and the resulting values were plotted to give the ternary phase diagram.

**Raw material preparation.** Olive pomace (OP) was purchased from Moulin Castelas (Baux-de-Provence, France), which uses a two-phase extraction olive oil unit. The proximate composition of the dried OP was determined using standard AOCS methods [42]. The ash content was determined by incineration at 500 °C. The moisture content was determined by oven drying the OP at 121 °C to constant weight. Kjeldahl method was used for total protein content analysis (standard method AOAC 928.08) using 6.25 as the nitrogen conversion factor [43]. Total oil content was determined according to standard procedure ISO 659.32 [44].

For the extraction of free phenolic compounds (FP), the dried OP was defatted using hexane for 60 min in an automatic Soxhlet extractor B-811 (BUCHI, Flawil, Switzerland) [45]. The defatted matrix was weighed (2.0 g) and FP were extracted according to Lopez-Martinez et al. (2009) protocol, with some modifications [46]. Extraction was performed with 20 mL of EtOH/H_2_O (70:30, *v*/*v*) at 8875 G for 10 min, using a T25 digital ULTRA-TURRAX (IKA, Staufen im Breisgau, Germany). Afterwards, the mixture was centrifuged at 8875 G for 5 min, the supernatant was removed, and solid residue extraction was repeated twice using the same procedure. The supernatants were pooled and concentrated under vacuum and the final volume was adjusted to 15 mL. The extraction of bound phenolic compounds (BP) was performed following the method described by Uribe et al. (2015), with some modifications [45]. The solid residue was hydrolysed in base (3 N NaOH, 20 mL) and agitated in an orbital shaker (88 min at 250 rpm). The hydrolysate was acidified to pH 3 (37% HCl). The liberated BP were extracted several times with ethyl acetate until the solvent became colourless. The pooled ethyl acetate extracts were evaporated to dryness under vacuum in a rotary evaporator (Büchi R-300, Flawil, Switzerland) at 40 °C. The total phenolic content was calculated by adding FP and BP, and was expressed as mg gallic acid equivalents (GAE)/100 g of dry matrix (DM) using the Folin-Ciocalteu assay. Total carbohydrates were calculated by difference (Tontisirin, 2003) [47] using Equation (1).
Total carbohydrates (%) = 100 − (moisture + protein + ash + oil + phenols)(1)

**Soxhlet extraction procedure.** The extractions were carried out with hexane (VWR, technical grade, Radnor, PA, USA), dry 2-MeOx (VWR, BHT stabilized, Radnor, PA, USA) and 2-MeOx 95.5%. Before each extraction, 2-MeOx was distilled to remove BHT. 2-MeOx 95.5% was prepared by mixing 2-MeOx with the corresponding amount of distilled water. Dried OP was extracted using a Soxhlet apparatus for 4.5 h. After extraction, the solvent was evaporated using a rotavapor (R-300 BÜCHI, Labortechnik AG, Flawil, Switzerland) at 40 °C and vacuum pressure of 50 mb. The potential solvent residues were removed by nitrogen flow. The crude extract was centrifuged (8875 G, 5 min) to remove any potential solid particles and the extraction yield was assessed on the liquid crude extract. The extraction yield was expressed as a percentage *w*/*w* of crude extracts per amount of matrix extracted, using the Equation (2).
Extraction yield (%) = Mass of crude extract/Mass of dried OP × 100(2)

Crude extracts were stored at 5 °C before analysis. The solid residue of the defatted OP was dried in a ventilated Biosec dehydrator (Tauro Essiccatori, Camisano vicentino, Italy) and stored at 5 °C. All analyses were performed in triplicate.

### 2.1. Crude extracts Analysis

**Fatty Acid Profile.** The acid-catalysed transesterification of OP crude extracts led to fatty acid methyl esters (FAMEs) [48]. In a glass tube, sulfuric acid in methanolic solution (1 mL, 5%, *v*/*v*) was added to the extracts (20–30 mg). The mixture was then heated in a heating block to facilitate transmethylation (90 min at 85 °C). Once at room temperature, NaCl solution (1.5 mL, 0.9% *m/v*) and *n*-hexane (1 mL) were added. The organic layer was collected and subjected to further analysis. Triheptadecanoin (C17:0; TAG) was used as an internal standard and FAME mix was used to calibrate the system. GC analyses were carried out in a gas chromatograph (Agilent 7820A, Santa Clara, CA, USA) coupled with a flame ionization detector (GC-FID) using a capillary column BD-EN14103 (30 m × 320 μm × 0.25 μm). The carrier gas (He) was set at 33 cm/s flow rate. The injector temperature was set at 250 °C working in split mode injection (2 μL, split ratio 1:20). The oven temperature ramp was the following: 50 °C for 1 min then a constant rate increase of 20 °C/min up to 180 °C, and then up to 230 °C at a rate of 2 °C/min. Once it reached 230 °C, the temperature was maintained for 10 min and the fatty acids were identified based on retention time and standards used for calibration.

**Neutral lipids**. The identification and relative quantification of neutral glycerides in OP crude extracts were performed via high-performance thin-layer chromatography (HPTLC) (ATS 5 automatic TLC sampler, ADC 2 automatic developing chamber, CAMAG 3 TLC scanning densitometer). Standard solutions of DL-α-palmitin (monoacyl glyceride, MAG), palmitic acid (free fatty acid, FFA), glyceryl 1,3-dipalmitate (diacyl glyceride, DAG), and glyceryl tripalmitate (triacylglyceride, TAG) were prepared in chloroform (0.2 mg/mL) and 10 mg of the sample lipid fraction was used for quantitation. A known volumes of extracts sample solutions (5 μL and 20 μL) were loaded onto 20 × 10 cm silica gel 60 F254 HPTLC plates. Neutral lipids were eluted with eluent B; a mixture of *n*-hexane/diethyl ether/glacial acetic acid in a ratio of 70:30:2 (*v*/*v*/*v*). Both plates were allowed to reach a height of 7 cm from the origin, then dried and dipped in a primuline dye reagent (10 mg of primuline, 160 mL of acetone, 40 mL of distilled water) for better visualisation of the lipid classes. Lipid standards were used to identify and quantify the lipid classes in crude extracts.

**Tocopherols and tocotrienols content, vitamin E activity, and squalene content.** Tocopherols and tocotrienols content were determined following ISO 9936 [49]. Vitamin E activity was calculated using Equation (3):Vitamin E activity = (*α*-tocopherol + 0.67 *α*-tocopherol acetate + 0.50 *β*-tocopherol + 0.10 *γ*-tocopherol + 0.03 *δ*-tocopherol)/10(3)
and expressed as mg of *α*-tocopherol equivalent (TE)/kg extract [50].

The squalene content was determined by gas chromatography, using an internal method. Analyses were conducted by ITERG analytical laboratory (Canejan, France).

**Carotenoid content**. Carotenoid contents were determined following the method of Mínguez-Mosquera et al. (1991). Briefly, samples of extracts were fully dissolved in cyclohexane at a known concentration. The absorbance of the solution was read at a wavelength of 470 nm. The content was quantified using Equation (4):Carotenoids (mg/kg of extract) = (A470 × 106)/(2000 × 100 × d)(4)
where A is the absorbance of the sample, 2000 is the extinction coefficient for carotenoids (lutein being a major component in the carotenoid fraction), and d is the spectrophotometer cell thickness (1 cm) [51].

**Total phenolic content (TPC) and 2,2-diphenylpicrylhydrazyl assay (DPPH).** Briefly, 1 g of crude extract was diluted in 1 mL of *n*-hexane and the phenolic compounds were extracted with 5 mL of methanol/water (80:20, *v*/*v*). The tube was vigorously shaken for 10 min, and, after centrifugation (8875 G, 10 min), the lower phase was collected. The upper layer was extracted two more times with the same procedure. Finally, the combined alcoholic phases were washed with a small volume of HPLC grade *n*-hexane and then transferred into a 15 mL volumetric flask [38,52]. Twenty microliters of appropriate dilutions of the extracts or gallic acid (standard) were placed into a 96-well microplate and 80 µL of a 7.5% Na_2_CO_3_ solution (*m*/*v*) was added and allowed to equilibrate at room temperature for 5 min. Thereafter, 100 µL of 1 N Folin-Ciocalteu reagent was added, and absorbance was read at 750 nm after incubation for 60 min. Results were calculated as gallic acid equivalents (GAE) per gram of crude extract, using distilled water as the blank [39]. Antioxidant activity was determined using the DPPH method [53], on a SPECTROstar omega microplate (BMG LABTECH GmBH, Ortenberg, Germany) reader equipped with UV-Vis spectrophotometer. Briefly, 50 µL of samples in the methanolic phase was allowed to react with 0.5 mM methanolic DPPH^•^ radical for 60 min and the absorbance was measured at 520 nm. Trolox was used as the standard for the calibration curve and methanol was used as the blank. Results are expressed as Trolox equivalents (TE) per gram of crude extract.

### 2.2. Chromatographic Analysis

**UPLC/DAD Solubility measurements and quantitative analysis of CA and C**. The concentration of the phenolic compounds in the saturated supernatants was assessed by UPLC/DAD analyses using an ACQUITY UPLC^®^ system (Waters, Milford, MA, USA) linked to a DAD diode array detector 200–800 nm (Waters, Milford, MA, USA). The separation was performed on an Acquity C18 BEH column (50 × 2.1 mm i.d., 1.7 mm). The solvents were (A) water/formic acid (99.5/0.5) and (B) acetonitrile. The gradient was linear, and the proportions of solvent B used were: 0–10 min, 1–20%, 10–12 min, 30%, 12–14 min, 100%. The injection volume was 1 μL and the column temperature was kept at 35 °C. The flow rate followed the same steps and was: 0.3 mL/min, 0.35 mL/min and 0.4 mL/min. Spectroscopic detection was performed in the range 200–600 nm with a resolution of 1.2 nm. The quantitative analysis of caffeic acid (CA) and *p*-coumaric acid (C) in the methanol: water extracts of the three crude extracts (hexane, dry 2-MeOx, 2-MeOx 95.5%) was carried out using the method described above. All analyses were performed in triplicate.

**HPLC-DAD Quantitative analysis of****HT, T, Oleacein (OE), Oleocanthal (OA).** HPLC-DAD analysis was used for the quantitative determination of major olive oil polyphenols in the three extracts as well as, the determination of the total amount of HT, T and their derivatives, after the hydrolysis procedure. In order to isolate the polar phenolic fraction from the crude extracts, a liquid-liquid extraction method assisted by centrifugation and ultrasound technique was applied (Determination of biophenols in olive oils by HPLC-COI-/T.20/Doc No 29) [54]. Specifically, 2.0 g of each sample and 1 mL of the internal standard solution (syringic acid) were added to a 10 mL test tube and vortexed for 30 s. Then, 5 mL of the methanol:water (80:20 *v*/*v*) were added and vortexed for a further minute. The mixture was sonicated in cleaning bath (15 min at room temp.) and centrifuged (5000 rpm, 25 min). Afterwards, an aliquot of the supernatant phase was taken, filtered and forwarded for HPLC analysis. Regarding the 2-MeOx-extracts samples, specific quantity was weighed, dissolved in methanol:water (1:1 *v*/*v*), filtered and forwarded for HPLC analysis as well. HT, T, OE and OA were quantified using the calibration curve method. On the other hand, the hydrolysis of secoiridoids to simple phenolic compounds and the determination of the total amount of HT, T and their derivatives was performed according to Brenes et al. (2001) [55]. Briefly, 25 mL of 2 M HCl was added to 1 g of extract in a 100 mL glass bottle. The mixture was vigorously agitated at ambient temperature for 6 h and forwarded for HPLC analysis. The total amount of HT, T and their derivatives was quantified by multiplying the quantities of HT and T phenyl alcohols with a correction factor (HT: 2.2 and T: 2.5), which corresponds to the difference of molecular weights of the coupled and free forms of HT and T [56]. HT and T were quantified using the calibration curve method. The HPLC-DAD analysis was performed according to analytical conditions referred to COI/T.20/Doc No 29 method (International Olive Council (IOC), Madrid, Spain, 2009) [54]. Analyses were carried out on a reversed-phase column (Spherisorb Discovery HS C18, 250 × 4.6 mm, 5 μm; Supelco) with aqueous orthophosphoric acid 0.2% (A) and methanol/acetonitrile (50:50 *v*/*v*) (B) as a mobile phase (flow rate of 1.0 mL/min) at room temperature. The injection volume was held constant at 20 μL. The applied gradient elution was as follows: 0 min, 96% A and 4% B; 40 min, 50% A and 50% B; 45 min, 40% A and 60% B; 60 min, 0% A and 100% B; 70 min, 0% A and 100% B; 72 min, 96% A and 4% B; 82 min, 96% A and 4% B. Chromatograms were monitored at 280 nm. All analyses were performed in triplicate.

## 3. Results and Discussion

**COSMO-RS Evaluation and solubility measurements.** COSMO-RS predicted the solubilities of some of the target components of olive oil and olive pomace (OP) in *n*-hexane, dry 2-MeOx and 2-MeOx 95.5%. The major constituents of olive oil (triglycerides, phospholipids, free fatty acids) were selected for the study, together with some minor compounds of high interest for cosmetic, pharmaceutical and food applications [57,58]. These minor compounds included tocopherols, sterols and carotenoids as well as phenolic compounds (alcohols, acids, rutosides, flavonoids, lignans and secoiridoids) [10]. Some of these results are presented in Table 1, expressed as log10 (x_solub_) the logarithm of the molar fraction of solute in the solvent, and showed that the investigated solutes had higher theoretical solubility in 2-MeOx (either dry or 95.5%), since the closer the value of log10 (x_solub_) to zero, the greater the predicted solubility of the solute. The solubility of triacylglycerides (TAG), which makes up approximately 99% of olive oil, was equivalent in the different solvents, as were the solubility of free fatty acids, hydrocarbons (squalene), triterpene compounds and carotenoids. Furthermore, sterols and lipid oxygenated compounds (e.g., tocopherols, aliphatic fatty alcohols) showed higher solubility in dry 2-MeOx and 2-MeOx 95.5% than in *n*-hexane, with much higher values for the polar lipids (e.g., phosphatidylcholine, phosphatidylinositol). All the different classes of phenolic compounds investigated showed much higher solubility in 2-MeOx (both dry and 95.5%) than in *n*-hexane, thanks to their greater polarity. The presence of phenolic glucoside derivatives has been extensively described [59,60], and the solubility of these compounds in *n*-hexane was negligible, whereas dry 2-MeOx and 2-MeOx 95.5% were again better solvents.

The solubilities of four phenolic compounds, hydroxytyrosol (HT), tyrosol (T), caffeic acid (CA) and *p*-coumaric acid (C), were determined experimentally in the three solvents studied (*n*-hexane, dry 2-MeOx and 2-MeOx 95.5%) at room temperature (25 °C) and compared with their COSMO-RS predicted solubilities (Table 1). In Figure 4, the molecular surface polarity distributions (σ-Profiles), and the molecular surface potential distributions (σ-Potentials) of the studied solvents and solutes are presented.

In accordance with the COSMO-RS predictions, the results showed that the four phenolic compounds had very high solubility in dry 2-MeOx and even higher in 2-MeOx 95.5%, whereas solubility was negligible in *n*-hexane due to its very low polarity. The presence of a small percentage of water in 2-MeOx 95.5% results in a strong increase in solubility for all four phenolic compounds. Solubility increases from dry 2-MeOx to 2-MeOx 95.5%: from 858.05 g/L to 2204.16 g/L for HT; from 379.38 g/L to 573.47 g/L for T; from 46.04 g/L to 133.69 g/L for CA; and, from 155.59 g/L to 237.11 g/L for C. Triolein (OOO), which is the major olive oil TAG, was completely miscible with all three solvents, as predicted by COSMO-RS calculations.

The ternary phase diagram at 55 °C of olive oil, 2-MeOx and water was investigated, as can be seen in Figure 5. The graph showed monophasic (orange) and biphasic (green) regions that are delimited by the experimentally determined points illustrated in yellow. The presence of a monophasic region corresponds to mixtures with moderate water concentrations, lower than 4.5% *w*/*w*. Moreover, as the percentage of oil increased, the amount of water that could be incorporated into the mixture progressively decreased.

The existence of this monophasic region allows 2-MeOx 95.5% to be considered a suitable solvent for olive oil extraction. In conclusion, higher extraction yields than in *n*-hexane are expected due to the similar solvent power of 2-MeOX (both dry and 95.5%) for all major olive oil compounds and the higher solubility of polar compounds, such as polar lipids and phenolic compounds.

Using 2-MeOx (both dry and 95.5%) COSMO-RS calculations and experimental studies predicted the possibility of obtaining an extract that is rich in bioactive compounds that have a potential beneficial effect on human health [61]. This first study represents an important initial step in determining whether 2-MeOx can successfully replace hexane in OP extraction.

**Proximate composition.** The studied OP, which was derived from a two-phase mill, initially contained a high percentage of water (approximately 67%), which is in line with values found in other OP samples that were recovered after a two-phase process [4]. The proximate composition was determined on the dried matrix (DM) and the percentage values are shown in Table 2.

The ash and protein contents were 4.18 and 6.64 g/100 g DM, respectively. The total oil content was 13.66 g/100 g DM. The total phenolic content was 2.24 g/100 g DM, which is in line with other values reported in the literature [13]. Of this, 1.97 g/100 g DM was ascribed to the class of free phenolic compounds (FP) and a smaller fraction (0.27 g/100 g DM) being ascribed to the class of bound phenolic compounds (BP). FP are small to medium-sized free polyphenols that are easily extracted with conventional solvents (water, ethanol, methanol) and have wide bioavailability due to favourable intestinal absorption. On the other hand, BP interact with other macromolecules (e.g., cellulose, lignin, proteins) via covalent bonds, and are extracted through acid or basic treatments that hydrolyse these bonds [62].

**Crude extract analyses.** The crude extracts obtained using the standard Soxhlet extractions in hexane, dry 2-MeOx and 2-MeOx 95.5% were characterized in terms of yield, chemical composition and quality (Table 3). The extraction yield of OP crude extracts was determined gravimetrically after 4.5 h extraction for the three solvents. As predicted by the COSMO-RS calculations, dry 2-MeOx and 2-MeOx 95.5% gave a higher extraction yield than hexane, presumably due to the extraction of further polar compounds. The highest extraction yield was obtained with dry 2-MeOx (15.68 g/100 g DM), followed by 2-MeOx 95.5% (14.10 g/100 g DM) and hexane (13.87 g/100 g DM). With 2-MeOx 95.5%, after solvent evaporation, the extract presented a small portion of solid, which was removed by centrifugation. This solid part could trap a portion of the liquid extract inside, and this feature potentially explains the higher yield obtained with dry 2-MeOx compared to 2-MeOx 95.5%.

The fatty acid profiles of the three extracts were fully comparable, with a higher content of oleic acid (C18:1), accounting for about 68% of the total. Palmitic acid (C16:0), linoleic acid (C18:2), stearic acid (C18:0) and linolenic acid (C18:3) were the other major compounds. All these compounds represent together about 98% of the total fatty acids, which is in agreement with literature data [4].

The three extracts had high mono-unsaturated fatty acid (Σ MUFA) and polyunsaturated fatty acid contents (Σ PUFA) contents. This resulted in them having saturated fatty acid content (Σ SFA) content that was low. Furthermore, the fatty acid profile of the three OP extracts were very similar to that of extra-virgin olive oils reported in the literature [57], which have a slightly higher oleic acid (C18:1) and lower linolenic acid (C18:2) contents. Diets rich in olive oil have been demonstrated to reduce total triglyceride level as well as total and low-density lipoprotein (LDL) due to the high-unsaturated fatty acid content, and this has positive effects on the cardiovascular system [63]. Nevertheless, it is essential to underline that the minor compounds in olive oil, especially polyphenols, play a key role in determining its quality and health benefits.

Quantification by HPTLC clearly showed that TAG was the principle neutral lipid compound present in all three of the analysed extracts, see Figure 6. On the other hand, free fatty acids (FFA), monoacylglycerides (MAG) and diacylglycerides (DAG) were not detected in any extracts. Traces of other compounds, probably polar compounds, were found in the dry 2-MeOx and 2-MeOx 95.5% extracts. These results are in agreement with literature that reports the TAG content of olive oil accounting for approximately 99% [64].

As shown in Table 4, α-tocopherol is the major tocopherol in all studied samples accounting for more than 96% of the total tocopherol and tocotrienol content. Furthermore, hexane induced the highest total content when compared to 2-MeOx (both dry and 95.5%). This tendency may be explained by the higher lipophilicity of hexane. Obviously, vitamin E activity followed the same tocopherol and tocotrienol contents trend. Although reported results showed a slightly lower concentration of tocopherols and tocotrienols in 2-MeOx, other studies on seed oils have shown that 2-MeOx extracts these compounds as efficiently as hexane [38,65].

Squalene is a triterpenic polyunsaturated hydrocarbon and showed to be present in high concentration in edible olive pomace oil, with a content ranging from 500 to 6000 mg/kg of oil [23]. When comparing squalene contents in the analysed samples, hexane exhibited the highest amount (5810 mg/kg extract) followed by dry 2-MeOx (4285 mg/kg extract) and 2-MeOx 95.5% (4033 mg/kg extract). Although during the refining processes, the squalene content has been documented to slightly decrease, all three crude extracts showed a concentration that is in agreement with the literature data.

On the other hand, the carotenoid contents showed an opposite trend leading to higher concentration in 2-MeOx (both dry and 95.5%) extracts compared to hexane.

**Phenolic Compounds.** Identification and quantification of the major extracts polyphenols were performed by HPLC-DAD according to COI/T.20/Doc No 29 method and by UPLC-DAD, see Table 5. The concentration of the main phenolic compounds in the extracts were found to increase in the following order: hexane, dry 2-MeOx and 2-MeOx 95.5% extracts. These results can be explained by the solubility of phenolic compounds being higher in dry 2-MeOx and even higher in 2-MeOx 95.5% in comparison to hexane, as shown in the preliminary solubility studies. Moreover, all three extracts were characterised by the presence of other bioactive compounds, including flavonoids (e.g., luteolin, apigenin), lignans (e.g., pinoresinol, acetoxypinoresinol) and other secoiridoids. Furthermore, 2-MeOx extracts (both dry and 95.5%) presented a very high concentration of major polyphenols, demonstrating that most olives phenolic compounds remain in the pomace during the olive oil production process, and that the hexane extraction does not allow their efficient recovery [13,66]. Oleacein and oleocanthal proved to be the two main phenolic compounds in terms of concentration in all the three extracts. These two compounds belong to the class of secoiridoids and over the past two decades have demonstrated several beneficial effects both in vitro and in vivo, including the protective effect against atherosclerosis, cancer and neurological diseases [67].

The total content of HT, T and their derivatives after acid hydrolysis was assessed. These values, which were in line with those previously discussed, showed an increase in the content of HT, T and derivatives, in the following order: hexane, dry 2-MeOx and 2-MeOx 95.5% extracts. Moreover, the total phenolic content (TPC) and the antioxidant activity (DPPH) were considerably higher in 2-MeOx (both dry and 95.5%) extracts than in hexane extract (Table 6). Several other studies have reported that 2-MeOx allows higher extraction of phenolic compounds from both plant (black cumin seeds, basil seeds, and soybean flakes) [38,65] and animal matrices (black soldier fly larvae) [39], resulting in extracts richer in polyphenols than those obtained with hexane and with greater antioxidant activities. In conclusion, the extracts obtained with 2-MeOx (both dry and 95.5%) have been shown to contain a considerably higher concentration of polyphenols than the extract obtained with hexane, as the apolar nature of hexane does not allow the efficient extraction of phenolic compounds to occur.

## 4. Conclusions

Nowadays, hexane is still the most commonly used solvent for the extraction of vegetable oils, including olive pomace oil, despite its harmfulness to human health and the environment has been well documented. In recent years, 2-MeOx has emerged as a credible bio-based alternative to hexane for the extraction of lipophilic natural products. In this study, the comparison between hexane and 2-MeOx (both dry and 95.5%) highlighted the potential of this bio-based solvent as a replacement for hexane in olive pomace extraction.

Thanks to theoretical studies using COSMO-RS and experimental evidence, 2-MeOx has proven to be an excellent solvent for the extraction of vegetable oils enriched with secondary metabolites, especially polyphenols, with interesting beneficial properties for human health.

The extraction yield was higher, with both dry 2-MeOx (15.68 ± 1.69 g/100 g DM) and 2-MeOx 95.5% (14.10 ± 0.34 g/100 g DM), than with hexane (13.87 ± 0.50 g/100 g DM), while the free fatty acid and neutral lipid profiles were shown to be fully comparable in the three extracts.

Furthermore, the total phenolic content (TPC) and the antioxidant activity (DPPH) of the extracts were significantly higher when using dry 2-MeOx (TPC 21989.22 μg GAE/g extract; DPPH 26307.66 μg TE/g extract) and 2-MeOx 95.5% (TPC 18487.30 μg GAE/g extract; DPPH 26681.75 μg TE/g extract), than hexane (TPC 862.51 GAE/g extract; DPPH 374.63 μg TE/g extract).

In particular, the concentrations of the main olive polyphenols Hydroxytyrosol (HT) and Tyrosol (T) in the three extracts studied were found to increase in the following order: hexane (HT 7.12 μg/g extract; T 24.11 μg/g extract), dry 2-MeOx (HT 1729.78 μg/g extract; T 771.46 μg/g extract), and 2-MeOx 95.5% (HT 1843.50 μg/g extract; T 893.69 μg/g extract).

The results presented in this study demonstrated that dry 2-MeOx and 2-MeOx 95.5% have the potential to replace hexane in olive pomace extraction. Polyphenols have been shown to be present in high concentrations in 2-MeOx crude extracts (both dry and 95.5%), possibly allowing the production of olive pomace oils rich in phenolic compounds. Furthermore, since during the chemical refining of the crude extracts, a portion of polyphenols is presumably lost, these important secondary metabolites could be recovered directly from the crude extracts, and used for nutraceutical, pharmaceutical and cosmetic applications.

Although 2-MeOx 95.5% may be a more cost-effective alternative to dry 2-MeOx, increased swelling of the matrix during extraction was noted, see Figure 7. This aspect, which has already been highlighted by Claux et al. (2021) in the extraction of soybean flakes, requires further investigation if the process has to be efficiently transferred to an industrial scale.

In conclusion, the demand for more sustainable and safer industrial extraction processes is constantly increasing and can be satisfied by fully applying the principles of green extraction and green chemistry [68,69,70]. Moreover, this alternative solvent can satisfy 11 of the 17 SDG (Sustainable Development Goals) adopted by all United Nations Member States in 2015 to provide a shared blueprint for peace and prosperity for people and the planet. The common links between the sustainable development goals (SDG) fulfilled by 2-MeOx and the principles of green chemistry and green extraction are presented in Figure 8, showing how 2-MeOx can certainly bring us one step further into an era of sustainable industrial solvent extraction.

## Figures and Tables

**Figure 1 foods-11-01357-f001:**
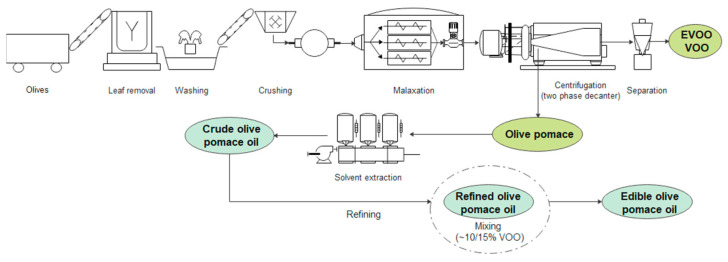
Olive oil and olive pomace oil production process.

**Figure 2 foods-11-01357-f002:**
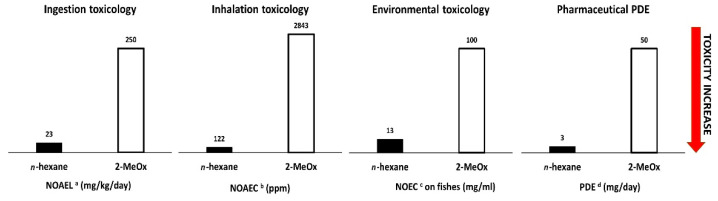
*n*-hexane and 2-MeOx Ingestion, Inhalation and Environmental toxicities and Pharmaceutical PDE comparison. ^a^ NOAEL—no-observed-adverse-effect level [28,29]; ^b^ NOAEC—no-observed-adverse-effect concentration [30,31]; ^c^ NOEC—no-observed-effect concentration [32,33]; ^d^ PDE—permitted daily exposure for an average human of 50 kg [34,35].

**Figure 3 foods-11-01357-f003:**
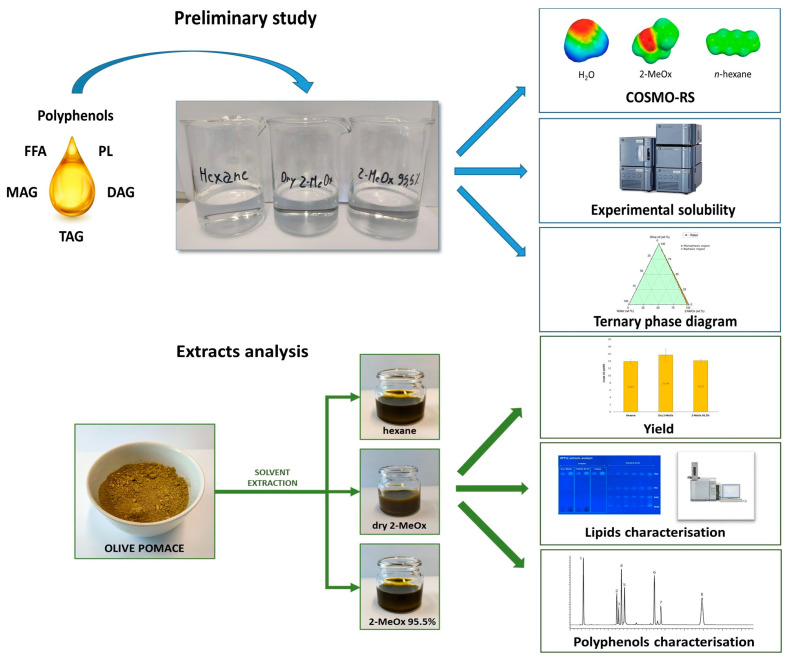
Study design overview.

**Figure 4 foods-11-01357-f004:**
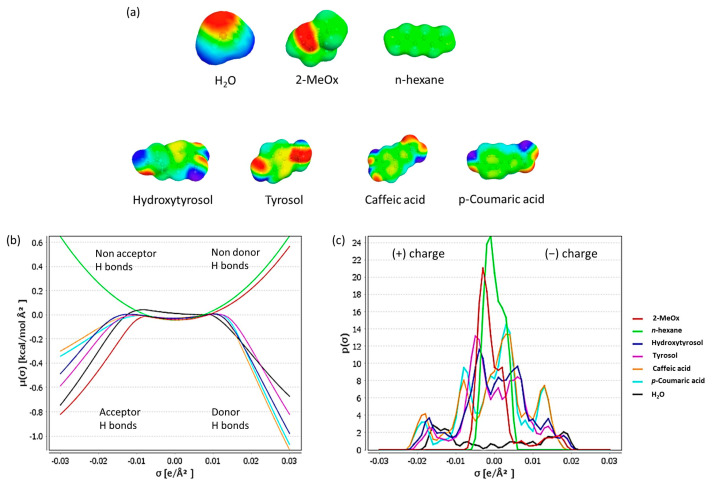
COSMO-RS theoretical solubility prediction: (**a**) solvents and solutes s-surface; (**b**) σ-Potential; (**c**) σ-Profile.

**Figure 5 foods-11-01357-f005:**
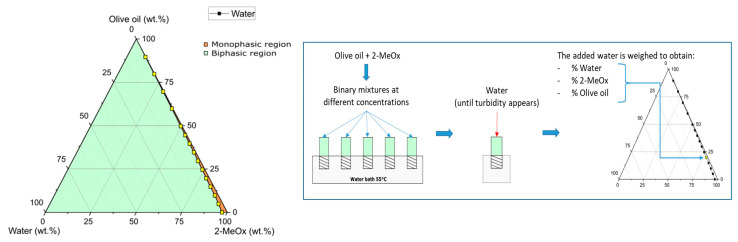
Ternary phase diagram at 55 °C of olive oil/2-MeOx/water.

**Figure 6 foods-11-01357-f006:**
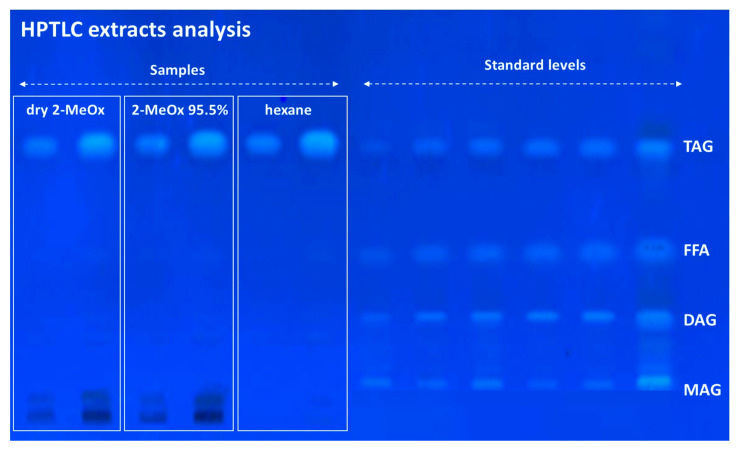
Neutral lipid analysis by HPTLC of OP extracts obtained with hexane, dry 2-MeOx and 2-MeOx 95.5%.

**Figure 7 foods-11-01357-f007:**
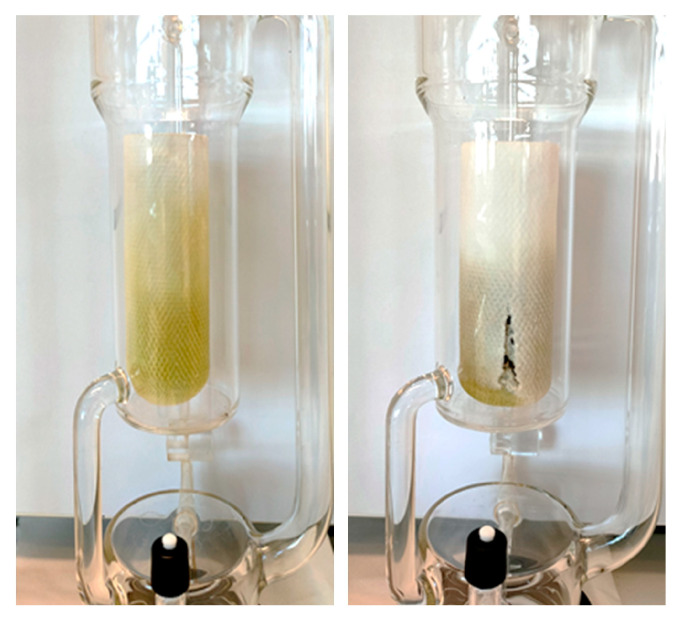
Dry 2-MeOx (left) and 2-MeOx 95.5% (right) Soxhlet extraction: cellulose cartridges initially filled with the same quantity of OP.

**Figure 8 foods-11-01357-f008:**
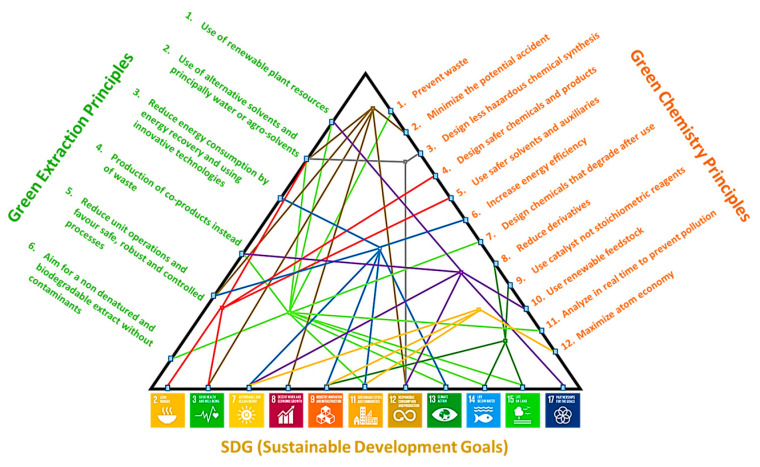
Correlation between SDG related to 2-MeOx and Green Chemistry and Green Extraction principles.

**Table 1 foods-11-01357-t001:** COSMO-RS predicted solubility and experimental solubility at room temperature (25 °C) of four phenolic compounds and Triolein (OOO) in n-hexane, dry 2-MeOx and 2-MeOx 95.5%.

		*n*-Hexane		Dry 2-MeOx		2-MeOx 95.5%	
		COSMO-RS (Log10 (x_solub_))	Experimental Solubility (g/L)	COSMO-RS (log10) (x_solub_))	Experimental Solubility (g/L)	COSMO-RS (Log10 (x_solub_))	Experimental Solubility (g/L)
Phenolic compounds	Hydroxytyrosol	−5.64	<0.01	0.00	858.05 ± 6.99	0.00	2204.16 ± 34.11
	Tyrosol	−4.93	0.01 ± 0.001	0.00	379.38 ± 9.56	0.00	573.47 ± 6.80
	Caffeic acid	−7.59	nd °	0.00	46.04 ± 0.49	0.00	133.69 ± 0.96
	*p*-Coumaric acid	−6.94	<0.01	0.00	155.59 ± 0.32	0.00	237.11 ± 0.29
TAG ^a^	Triolein (OOO)	0.00	∞	0.00	∞	0.00	∞

^a^ TAG, Triacylglyceride; ° nd, not detected, under detection limit for the methodology and equipment used. Mean ± standard deviation of determinations (*n* = 3).

**Table 2 foods-11-01357-t002:** OP proximate composition.

	Content
Ash (g/100 g DM)	4.18 ± 0.66
Oil (g/100 g DM)	13.66 ± 1.01
Protein (g/100 g DM)	6.64 ± 0.39
Total phenolic content (g GAE/100 g DM) Free phenolic compounds Bound phenolic compounds	2.24 ± 0.10 1.97 ± 0.06 0.27 ± 0.04
Carbohydrates (g/100 g DM)	73.28 ± 2.14

DM, dried matrix.

**Table 3 foods-11-01357-t003:** Extraction yield and fatty acid profile of OP extracts obtained with hexane, dry 2-MeOx and 2-MeOx 95.5%.

	Hexane	Dry 2-MeOx	2-MeOx 95.5%
Extraction yield (g/100 g DM)	13.87 ± 0.50	15.68 ± 1.69	14.10 ± 0.34
Fatty acid profile (relative %)			
C14	traces	traces	traces
C14:1	traces	traces	traces
C16	15.20 ± 0.09	15.08 ± 0.02	15.19 ± 0.13
C16:1	0.80 ± 0.46	1.07 ± 0.01	1.00 ± 0.07
C18	2.72 ± 0.02	2.66 ± 0.03	2.77 ± 0.08
C18:1 *n*-9	68.05 ± 0.28	67.65 ± 0.02	67.43 ± 0.20
C18:2 *n*-6 cis	11.77 ± 0.09	11.96 ± 0.09	11.34 ± 0.66
C18:3 *n*-3	0.79 ± 0.02	0.91 ± 0.01	0.80 ± 0.10
C20	0.29 ± 0.01	0.27 ± 0.01	0.28 ± 0.01
C20:1 *n*-9	0.33 ± 0.01	0.34 ± 0.11	1.13 ± 0.82
Others	0.05 ± 0.01	0.07 ± 0.01	0.05 ± 0.01
Σ SFA ^a^	18.20 ± 0.12	18.00 ± 0.02	18.24 ± 0.21
Σ MUFA ^b^	69.19 ± 0.21	69.06 ± 0.10	69.57 ± 0.55
Σ PUFA ^c^	12.56 ± 0.11	12.87 ± 0.10	12.14 ± 0.76
C18:1/C18:2	5.78 ± 0.03	5.66 ± 0.04	5.95 ± 0.33

^a^ Saturated fatty acids; ^b^ Monounsaturated fatty acids; ^c^ Poly-unsaturated fatty acids. Mean ± standard deviation of determinations (*n* = 3).

**Table 4 foods-11-01357-t004:** Tocopherol, tocotrienol, squalene and carotenoids content in OP extracts obtained with hexane, dry 2-MeOx and 2-MeOx 95.5%.

	Hexane	Dry 2-MeOx	2-MeOx 95.5%
Tocopherol and Tocotrienol content (mg/kg extract)			
*α*-tocopherol acetate	<5	<5	<5
*α*-tocopherol	288	234	201
*β*-tocopherol	4	3	3
*γ*-tocopherol	7	6	5
*δ*-tocopherol	<2	<2	<2
*α*-tocotrienol	<2	<2	<2
*β*-tocotrienol	<2	<2	<2
*γ*-Tocotrienol	<2	<2	<2
*δ*-Tocotrienol	<2	<2	<2
Total	300 ± 45	243 ± 36	209 ± 31
Vitamine E activity (mg *α*-TE ^a^/kg extract)	291	236	203
Squalene and carotenoids content (mg/kg extract)			
Squalene	5810	4285	4033
Carotenoids	11.97 ± 0.32	134.39 ± 7.37	149.00 ± 2.50

^a^ α-Tocopherol equivalent.

**Table 5 foods-11-01357-t005:** Phenolic compounds (mg/Kg extract) quantification in OP extracts obtained with hexane, dry 2-MeOx and 2-MeOx 95.5%.

	Hexane	Dry 2-MeOx	2-MeOx 95.5%
Hydroxytyrosol ^a^	7.12 ± 0.84	1729.78 ± 34.30	1843.50 ± 22.77
Tyrosol ^a^	24.11 ± 2.14	771.46 ± 14.34	893.69 ± 20.33
Oleacein ^a^	140.56 ± 14.17	6144.46 ± 11.34	7698.91 ± 34.07
Oleocanthal ^a^	766.39 ± 10.34	3596.98 ± 46.55	3425.47 ± 22.34
Caffeic acid ^b^	nd °	287.10 ± 4.34	310.43 ± 7.07
*p*-Coumaric acid ^b^	nd °	698.42 ± 7.27	808.36 ± 14.59

^a^ HPLC-DAD analysis; ^b^ UPLC-DAD analysis; ° nd, not detected. Mean ± standard deviation of determinations (*n* = 3 per each sample replicate).

**Table 6 foods-11-01357-t006:** Determination of total amount of HT, T and their derivatives, after hydrolysis of extracts products, and total phenolic content and antioxidant activity of extracts.

	Hexane	Dry 2-MeOx	2-MeOx 95.5%
Total HT ^a^ (mg/kg extract product)	194.42 ± 14.10	5585.06 ± 34.34	7293.06 ± 24.06
Total T ^a^ (mg/kg extract product)	523.69 ± 24.34	4934.31 ± 50.06	4618.25 ± 40.23
Total HT, T and their derivates ^a,b^ (mg/kg extract product)	1736 ± 45.16	24622 ± 99.17	27590 ± 75.54
Total phenolic content (mg GAE/kg extract)	862.51 ± 17.99	21989.22 ± 834.62	18487.30 ± 255.24
Antioxidant activity (mg TE/kg extract)	374.63 ± 79.45	26307.66 ± 849.86	26681.75 ± 452.69

^a^ after acid hydrolysis in extracts products; ^b^ calculated by multiplying the quantities of HT and T phenyl alcohols, obtained by the hydrolysis process, with a correction factor (HT: 2.2 and T: 2.5). Mean ± standard deviation of determinations (*n* = 3 per each sample replicate).

## Data Availability

Data is contained within the article and also available on request.

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
