# Peer review of "Higher Yield and Polyphenol Content in Olive Pomace Extracts Using 2-Methyloxolane as Bio-Based Solvent"

_foods, 2022, doi:10.3390/foods11091357_

Round 1

Reviewer 1 Report

the writing needs a review of the numbering and references of the tables and figures. The references to the quotations must also be corrected; an error message often appears in the text instead of the correct reference.

In table 1 in line 362 the value 0.01 + 0.0003 is reported; the figures shown in the measure are inconsistent with those of the error.

Author Response

Reviewer 1:

Comments and Suggestions for Authors

the writing needs a review of the numbering and references of the tables and figures. The references to the quotations must also be corrected; an error message often appears in the text instead of the correct reference.

All the authors warmly acknowledge the Reviewer for the positive assessment and the useful suggestions.

All the amendments have been carefully corrected.

In table 1 in line 362 the value 0.01 + 0.0003 is reported; the figures shown in the measure are inconsistent with those of the error.

In table 1 (line 362), the standard deviation has been corrected to 0.01 ± 0.001. 

Reviewer 2 Report

Please correct

'Error! Reference source not 321 found.'

 throughout the paper....

We detect 30% of the repetition rate, however, the research was conducted comprehensively, and the authors clearly described the materials and methods and presented their results. I would certainly praise both the setting up of the experiment and the innovation.

All in all a very good scientific paper and I recommend minor corrections.

Author Response

Referee 2:

Comments and Suggestions for Authors

Please correct

'Error! Reference source not 321 found.'

 throughout the paper....

The authors warmly acknowledge the Reviewer for the positive assessment and the useful comments.

All the amendments have been carefully corrected.

We detect 30% of the repetition rate, however, the research was conducted comprehensively, and the authors clearly described the materials and methods and presented their results. I would certainly praise both the setting up of the experiment and the innovation.

The manuscript was thoroughly revised by rephrasing several sentences reported in previous papers.

All in all a very good scientific paper and I recommend minor corrections.

Reviewer 3 Report

The manuscript describes a study regarding the possible replacement of hexane by 2-MeOx in the extraction of olive pomace. The study includes theoretical and practical aspects, through the use of appropriate and current techniques. The research topic is important and may have practical implications related to ESG practices.

Please, give details on how Figure 8 was obtained.   

Minor corrections:

lines 392 to 396. Bound phenolics do not have high molecular weight (e.g. ferulic acid is a bound phenolic). Rephrase the sentence.

All over the manuscript there are Figures and Tables quoted as "Error! Reference source not found". Check for correction. 

Author Response

Referee 3:

Comments and Suggestions for Authors

The manuscript describes a study regarding the possible replacement of hexane by 2-MeOx in the extraction of olive pomace. The study includes theoretical and practical aspects, through the use of appropriate and current techniques. The research topic is important and may have practical implications related to ESG practices.

The authors warmly acknowledge the Reviewer for the positive assessment and the useful comments.

All the amendments have been carefully corrected.

Please, give details on how Figure 8 was obtained.   

Figure 8 was obtained by connecting the common points of the principles of green extraction and green chemistry, with the SDGs (Sustainable and Development goals) considered to be satisfied by the use of 2-methyloxolane (2-MeOx). Technically, the Figure 8 was made using Power Point.

Minor corrections:

lines 392 to 396. Bound phenolics do not have high molecular weight (e.g. ferulic acid is a bound phenolic). Rephrase the sentence.

All over the manuscript there are Figures and Tables quoted as "Error! Reference source not found". Check for correction. 

The highlighted sentences have been rephrased and the manuscript was thoroughly revised by rephrasing several sentences reported in previous papers.
